# The First Few Tokens Are All You Need: An Efficient and Effective *Unsupervised Prefix* Fine-Tuning Method for Reasoning Models

**Ke Ji**[*,1,2] , **Jiahao Xu**[*,2] , **Tian Liang**[*,2] , **Qiuzhi Liu**[*,2] , **Zhiwei He**[2] , **Xingyu Chen**[2] ,
**Xiaoyuan Liu**[2] , **Zhijie Wang**[2] , **Junying Chen**[1] , **Benyou Wang**[†,1] ,
**Zhaopeng Tu**[†,2] , **Haitao Mi**[2] , **and Dong Yu**[2]

[1]The Chinese University of Hong Kong, Shenzhen
[2]Tencent

## Abstract

Improving the reasoning capabilities of large language models (LLMs) typically requires supervised fine-tuning with labeled data or computationally expensive sampling. We introduce Unsupervised Prefix Fine-Tuning (UPFT), which leverages the observation of Prefix Self-Consistency – the shared initial reasoning steps across diverse solution trajectories – to enhance LLM reasoning efficiency. By training exclusively on the initial prefix substrings (as few as 8 tokens), UPFT removes the need for labeled data or exhaustive sampling. Experiments on reasoning benchmarks show that UPFT matches the performance of supervised methods such as Rejection Sampling Fine-Tuning, while reducing training time by 75% and sampling cost by 99%. Further analysis reveals that errors tend to appear in later stages of the reasoning process and that prefix-based training preserves the model's structural knowledge. This work demonstrates how minimal unsupervised fine-tuning can unlock substantial reasoning gains in LLMs, offering a scalable and resource-efficient alternative to conventional approaches.

## 1 Introduction

Large language models (LLMs) have demonstrated remarkable performance across a wide range of natural language understanding and generation tasks, primarily due to large-scale pre-training and subsequent instruction fine-tuning on high-quality datasets [16, 27]. Despite these successes, enabling LLMs to exhibit systematic reasoning capabilities remains a challenging endeavor [29, 4, 8, 20]. In multiple domains—from mathematical problem solving to logical and commonsense reasoning—models often rely on large amounts of human-annotated data or extensive sampling-and-filtering pipelines to achieve high accuracy.

Recent inquiry has introduced approaches such as Rejection Sampling Fine-Tuning [33] (RFT) and Self-Taught Reasoner [34] (STaR) to leverage model-generated solutions for iterative **self-improvement**, often requiring multiple candidate responses and subsequent filtering or verification steps. While these methods can yield impressive gains, they are time-consuming, resource-intensive, and assume ready access to correct targets or verification mechanisms — particularly challenging when no reliable ground-truth is available.

---

[*]Equal Contribution. The work was done when Ke Ji, Zhiwei He, Xingyu Chen, Xiaoyuan Liu, and Zhijie Wang were interning at Tencent.
[†]Correspondence to: Benyou Wang <wangbenyou@cuhk.edu.cn> and Zhaopeng Tu <zptu@tencent.com>.

39th Conference on Neural Information Processing Systems (NeurIPS 2025).

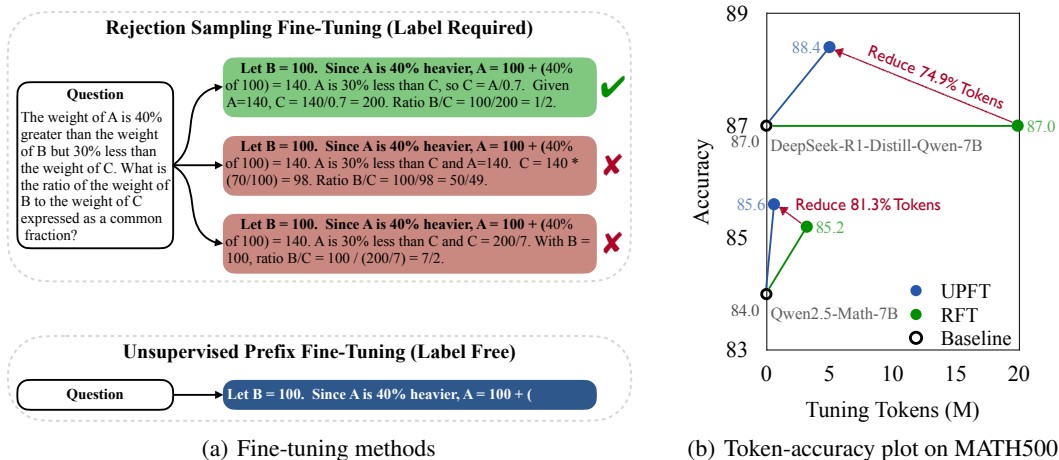

(a) Fine-tuning methods      (b) Token-accuracy plot on MATH500

Figure 1: (a): Conventional Rejection Sampling Fine-Tuning (RFT) method (upper panel) involves generating multiple responses to a given question and then applying posterior filtering to discard trajectories that lead to incorrect answers. Finally, the correct trajectory is used for final training. In contrast, the proposed UPFT method (bottom panel) requires only prefix minimal initial tokens of a single generated sample, eliminating the need for labeled data or rejection sampling. (b): Our proposed UPFT matches the performance of supervised RFT, while reduces tuning cost by 75+%.

In this paper, we propose an unsupervised fine-tuning method that requires only a single pass of model-generated responses per question, coupled with prefix-based fine-tuning. Our key insight is that different solution paths often share a common initial reasoning phase, which we call "Prefix Self-Consistency". By fine-tuning on these minimal prefixes (as few as 8 tokens), we effectively guide the model's inherent reasoning structures toward more systematic solution strategies while avoiding the complexity and cost of large-scale or iterative filtering. Moreover, we preserve the model's overall problem-solving format through a small proportion of full-token fine-tuning experiments, ensuring the model does not lose its length generalization and instruction-following abilities.

We conduct comprehensive experiments across four training corpora and evaluate our method on four widely used reasoning benchmarks. UPFT demonstrates exceptional data efficiency and flexibility, outperforming conventional full-token fine-tuning in an unsupervised sampling setting. Furthermore, UPFT achieves performance competitive with the widely used RFT method, which relies on labeled verification or large-scale rejection sampling, while requiring only 25% of the training time and 1% of the sampling time. Our approach can be easily adapted to various datasets, tasks, and LLM architectures, highlighting its flexibility and universality for developing robust problem-solving capabilities in large language models.

Our main contributions are as follows:

1. We identify Prefix Self-Consistency as a critical phenomenon, showing that early reasoning steps are highly consistent across trajectories, enabling efficient self-improvement learning.

2. We propose an unsupervised fine-tuning method UPFT that leverages only prefix substrings (i.e., minimal initial tokens of model-generated responses), eliminating the need for labeled data or rejection sampling.

3. We conduct comprehensive empirical validation to demonstrate that UPFT exhibits exceptional data efficiency and versatility, outperforming vanilla full-token fine-tuning in unsupervised settings and achieving competitive performance with RFT while drastically reducing sampling and tuning overhead.

## 2 Prefix Self-Consistency

A central observation of this work is that different solution trajectories for the same question often share a common *initial* reasoning phase, even if their later steps diverge substantially. We term this phenomenon **prefix self-consistency**. In other words, when an LLM generates multiple solutions for a single question, the opening tokens – typically restatements of the problem or the setup of its

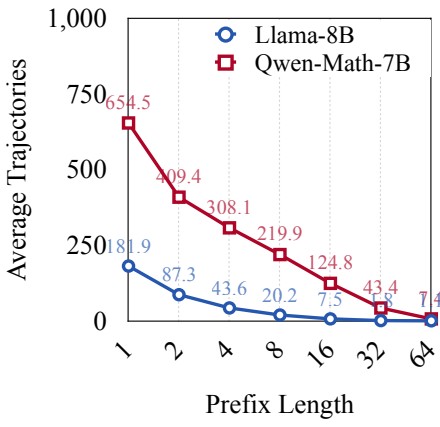

(a) Average number of covered trajectories.

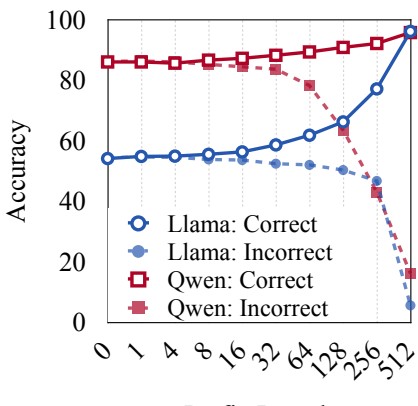

(b) Success rate of trajectory prefixes.

Figure 2: An empirical investigation of prefix self-consistency. We investigate (a) the average number of trajectories covered by prefixes at different lengths, and (b) the success rate of 32 rollout samplings from prefixes for both correct and incorrect trajectories.

initial logical steps – tend to be highly consistent across these separate responses. In this section, we identify two key characteristics of reasoning prefixes to demonstrate the existence and plausibility of prefix self-consistency. We report results on 500 questions randomly sampled from the PRM training dataset.

**Early reasoning steps are highly consistent across reasoning trajectories.** We sampled 1,000 trajectories for each question instance (using a temperature of 0.6) and calculated the average trajectories per prefix of different lengths, as shown in Figure 2(a). The results confirm that reasoning processes exhibit strong prefix self-consistency, with a remarkably high degree of prefix overlap among multiple trajectories for both models. As we increase $t$ (i.e., longer prefixes), the average number of samples per prefix pattern decreases, yet both models maintain consistent patterns well beyond the very first tokens. Notably, the math-specialized Qwen-Math-7B-Instruct preserves shared prefixes more consistently than the general-purpose Llama-3.1-8B-Instruct, as evidenced by the higher values in all cases. This suggests that Qwen-Math-7B's generation process is more tightly anchored in its initial reasoning steps, whereas Llama-3.1-8B introduces more prefix variability. An example is shown in Appendix D.

**Errors predominantly occur in later reasoning steps.** To empirically validate the plausibility of the reasoning prefix, we conducted 32 rollout samplings for each token in both a correct and an incorrect trajectory for each question. Figure 2(b) illustrates the success rate of these rollout samplings across both correct and incorrect trajectories. Two key observations can be made:

- *Incorrect trajectories diverge significantly from correct ones in later reasoning steps.* The rollout success rate for correct trajectories steadily rises as $t$ grows for both models. For example, with Llama, the success rate starts at 54.2% at $t = 0$ and reaches 96.2% by $t = 512$. This trend is expected: sampling from the later tokens of a correct trajectory leverages more accurate contextual information, increasing the likelihood of staying on a correct path. In contrast, the rollout success rate for incorrect trajectories declines substantially as $t$ increases, indicating that once an incorrect path is taken, recovering through rollout sampling becomes far less likely. This contrast highlights that errors tend to occur in the later stages of generation.

- *Initial reasoning prefixes exhibit self-consistency.* At the earliest token positions ($t \leq 16$), the rollout success rates for both correct and incorrect trajectories are strikingly similar. For Qwen-Math-7B at $t = 4$, the success rates are 85.7% and 86.1% for correct and incorrect trajectories, respectively. This near-identical performance in early steps suggests that the initial tokens generated – whether they lead to a correct or incorrect final answer – are statistically indistinguishable in terms of leading to a correct result when used as a starting point for rollout sampling.

Overall, these results indicate that while mistakes in reasoning are more prone to appear and accumulate in later steps, the initial reasoning prefixes generated by LLMs exhibit a considerable degree of self-consistency. This consistency is reflected in the similar early-stage rollout success rates across both correct and incorrect trajectories. Consequently, enhancing the accuracy and robustness of these early reasoning steps may be a key factor in improving the overall reliability of reasoning tasks.

## 3  Methodology

Based on the aforementioned observation, we firstly provide a Bayesian explanation in Section 3.1, identify the coverage and accuracy of **Prefix Self-Consistency** in Section 3.2, and propose our UPFT in Section 3.3.

### 3.1  Understanding Coverage and Accuracy from A Bayesian Perspective

We demonstrate that the SFT process can be naturally interpreted within a Bayesian framework. Consider a dataset $(x, y) \sim \mathcal{D}$, where each $x$ represents an input and $y$ is the corresponding ground-truth answer. For each input $x$, we conduct the reject sampling process of SFT aims to maximize the following objective:

$$\mathbb{E}_{\substack{(x,y)\sim\mathcal{D},\ r^{(k)}\sim p(\cdot|x;\theta) \\ r\in_R \mathbb{1}(r^{(k)},y)}} \log p(r|x) \tag{1}$$

where $r^{(k)} \sim p(\cdot|x; \theta)$ denotes the generated reasoning trace, and $r \in_R \mathbb{1}(r^{(k)}, y)$ denotes the rejection sampling, which only picks the reasoning trace $r$ whose final answer is $y$.

From a probabilistic perspective, we can decompose the conditional probability $p(y|x)$ by marginalizing over all possible reasoning traces $r$ that could lead to the answer, and derive the following lower bound:

$$\log p(y|x) = \log \sum_r p(y, r|x) \tag{2}$$

$$= \log \sum_r p(y|r, x)p(r|x) \tag{3}$$

$$\geq \sum_r p(r|x) \log p(y|r, x) \tag{4}$$

$$= \mathbb{E}_{r\sim p(\cdot|x)} \log p(y|r, x) \tag{5}$$

The correct answer $\log p(y|x)$ is lower-bounded by:

- **Coverage**: The expectation $\mathbb{E}_{r\sim p(\cdot|x)}[\cdot]$ is with respect to the distribution of reasoning trace $p(r|x)$. The term $p(r|x)$ denotes a prior probability of reasoning trace $r$ given input $x$, which denotes the prefix coverage of the entire reasoning trace space. This corresponds to the average trajectory covered by prefixes in Section 2 indicated by Figure 2(a).

- **Accuracy**: Term $p(y|r, x)$ within the expectation can be viewed as the likelihood of the answer $y$ being correct given a specific reasoning trace $r$, which is the accuracy. This is also observed by our previous observation in Section 2 in Figure 2(b).

### 3.2  Prefix Coverage and Accuracy

To incorporate the concept of prefix spans, we utilize the chain rule for conditional probabilities. We can decompose the probability of the full reasoning trace $r$ given input $x$ as:

$$p(r|x) = p(r_{<t}, r_{\geq t}|x) = p(r_{<t}|x)p(r_{\geq t}|r_{<t}, x)$$

where $r_{<t}$ denotes the prefix of $r$ before time step $t$. Substituting this decomposition of $p(r|x)$ into the original lower bound from Equation (4), we get:

$$\log p(y|x) \geq \sum_r p(r|x) \log p(y|r, x)$$

$$= \sum_r p(r_{<t}|x) p(r_{\geq t}|r_{<t}, x) \log p(y|r_{<t}, r_{\geq t}, x)$$

$$= \sum_{r_{<t}} \left[ p(r_{<t}|x) \sum_{r_{\geq t}} p(r_{\geq t}|r_{<t}, x) \log p(y|r_{<t}, r_{\geq t}, x) \right]$$

$$= \sum_{r_{<t}} p(r_{<t}|x) \left[ \sum_{r_{\geq t}} p(r_{\geq t}|r_{<t}, x) \log p(y|r, x) \right]$$

$$= \sum_{r_{<t}} p(r_{<t}|x) L(r_{<t}, x)$$

$$= \mathbb{E}_{r_{<t} \sim p(\cdot|x)}[L(r_{<t}, x)] \tag{6}$$

where,

$$L(r_{<t}, x) = \sum_{r_{\geq t}} p(r_{\geq t}|r_{<t}, x) \log p(y|r_{<t}, r_{\geq t}, x)$$

$$= \mathbb{E}_{r_{\geq t} \sim p(\cdot|r_{<t}, x)}[\log p(y|r_{<t}, r_{\geq t}, x)] \tag{7}$$

Equation (6) presents the original lower bound of Equation (5) in terms of prefix spans of the reasoning trace. It also states the importance of coverage and accuracy of the prefix span $r_{<t}$:

- **Prefix Coverage**: The outer expectation in Equation (6) $\mathbb{E}_{r_{<t} \sim p(\cdot|x)}[\cdot]$ is with respect to the distribution of prefixes $p(r_{<t}|x)$. The term $p(r_{<t}|x)$ represents the prior probability of the model generating a prefix reasoning trace $r_{<t}$ given the input $x$. This denotes the coverage of prefix reasoning traces.

- **Prefix Accuracy**: For each prefix $r_{<t}$, the term $L(r_{<t}, x)$ in Equation (7) is the conditional lower bound given the prefix $r_{<t}$. It is the expected log-likelihood of the answer $y$ given that the reasoning process starts with prefix $r_{<t}$, averaging over all possible suffixes $r_{\geq t}$ that can follow $r_{<t}$. This can be viewed as the expected accuracy when reasoning is conditioned to start with the prefix $r_{<t}$.

So far, we have mathematically derived a representation of the lower bound in terms of prefix spans for both prefix coverage and prefix accuracy. To achieve superior performance through prefix fine-tuning, a specific prefix length $t$ is required to trade off both prefix coverage and prefix accuracy terms.

### 3.3  Our Method: Unsupervised Prefix Fine-Tuning

Motivated by previous theoretical justification, we develop our method Unsupervised Prefix Fine-Tuning (UPFT), which is apt to maximize the coverage of the reasoning trace while maintaining a relatively high accuracy by only learning from the proper prefix of the reasoning trace. This strategy shares several advantages. First, *UPFT exhibits enhanced coverage of all possible correct reasoning traces.* As demonstrated in Section 2 and Section 3.1, the prefix of the reasoning traces represents a set of reasoning traces with the identical prefix, which increases the coverage term of Equation (5)'s lower bound. Second, *by solely decoding a certain length of the prefix, UPFT inherently enjoys improved computational efficiency.* In contrast, conventional Reasoning From Traces (RFT) methods necessitate rejecting sampling over entire reasoning traces. This approach incurs a significant inference burden, particularly in scenarios where generating valid full reasoning traces from the base model proves challenging.

Our strategy consists of the following steps:

- Given a training set $(x, y) \in \mathcal{D}$, we only decode a prefix span $r_{<t}$ of reasoning trace from the base model $r_{<t} \sim p(\cdot|x; \theta)$;
- We conduct the SFT learning on the prefix spans of the reasoning trace $r_{<t}$ with NLL objective;

**Structure Tuning**   To avoid the catastrophic forgetting [19], we jointly conduct the prefix tuning and full trace tuning by a data split ratio $p\%$, which yields a prefix dataset $\mathcal{D}_p$ and full trace dataset $\mathcal{D}_f$. We utilize a specialized task template, as shown in Appendix A, for prefix tuning to enhance the model's ability to learn effectively from prefix-specific patterns. Note that the full trace tuning does not require correctness verification as well. In summary, the overall objective is a combination of UPFT and unsupervised SFT by maximizing the following objective:

$$\mathbb{E}_{\substack{x\sim\mathcal{D}_p \\ r_{<t}\sim p(\cdot|x;\theta)}} \left[\log p(r_{<t}|x;\theta)\right] + \mathbb{E}_{\substack{x\sim\mathcal{D}_f \\ r\sim p(\cdot|x;\theta)}} \left[\log p(r|x;\theta)\right] \tag{8}$$

Notably, compared to conventional RFT objective, our method does not require any rejection sampling process denoted by $\mathbb{1}(r^{(k)}, y)$ in Equation (1). Consequently, our method is naturally suitable for learning from unproven questions with extreme data efficiency.

## 4   Experiment

In this experiment, we compare UPFT with traditional supervised methods such as SFT and RFT in both supervised and unsupervised sampling settings. We also evaluate its scalability by varying the self-training corpora and backbone models.

### 4.1   Experiments Setup

**Backbone LLMs**   For our experiments, we selected three representative open-source backbone models: the general-purpose Llama-3.1-8B-Instruct [4], the math-specialized Qwen2.5-Math-7B-Instruct [29] optimized for mathematical tasks, and the long-reasoning DeepSeek-R1-Distill-Qwen-7B [8], distilled from DeepSeek-R1.

**Fine-Tuning**   We used four datasets to generate self-training data for fine-tuning:

1. **PRM** (12K instances; 15): This dataset includes 4.5K MATH [10] test problems, which are drawn from the PRM800K training set.
2. **OMI2** (600K instances; 26): This is a subset of 600K unique questions extracted from the OpenMathInstruct2 math-instruction tuning dataset.
3. **LIMO** (819 instances; 31): A high-quality training dataset specifically focused on challenging problems.
4. **U-Hard** (100K questions): This dataset, introduced in this work, is designed to explore the potential of our method in an unsupervised setting.

U-Hard was curated through an extensive collection of questions from publicly available online sources. Adhering to the Omni-Math approach [6], we labeled the collected data by difficulty and subsequently filtered out lower-difficulty questions. This process resulted in a dataset composed exclusively of challenging questions, thereby ensuring U-Hard's practical relevance to real-world scenarios. To ensure a fair comparison, UPFT employs the same hyperparameters as conventional SFT when fine-tuning models.Details about hyperparameters is shown in Appendix B. During the inference stage, we adopt a prompted zero-shot setup and use standard greedy decoding, wherein models are directed to answer each question using natural language instructions without any accompanying contextual demonstrations.

**Benchmarks**   We evaluated the performance of our method and the baseline methods on four widely used benchmarks: **GSM8K** [3], **MATH500** [10], **AIME24** [18], and **GPQA Diamond** [24].

**Tuning Scenarios**   We employed two experimental settings:

- **Unsupervised sampling**: we took one sample per question without any posterior filtering.
- **Supervised sampling**: Following RFT [33], we sampled the responses for each question 16 times and used the ground truth answers to select a correct response.

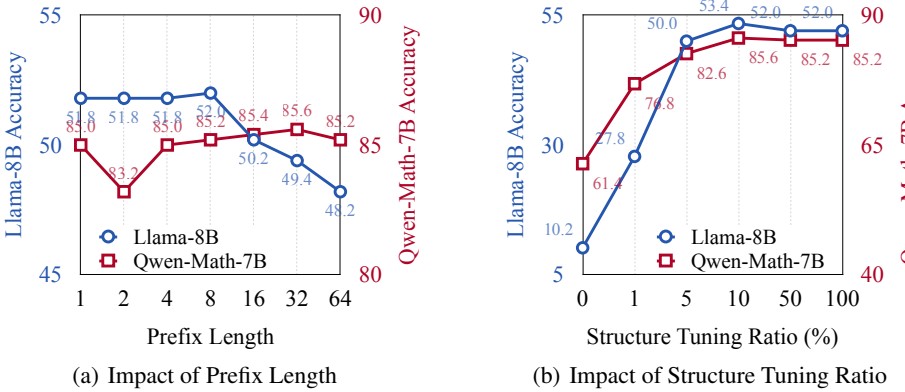

Figure 3: Impact of (a) prefix length and (b) structure tuning ratio on reasoning accuracy.

## 4.2 Ablation Study

In this section, we evaluated the impact of two hyperparameters of UPFT on the PRM-12K dataset using Llama-3.1-8B-Instruct and Qwen2.5-Math-7B-Instruct in the unsupervised setting and report the results on the MATH500 test set.

**Impact of Prefix Length**  Figure 3(a) illustrates how prefix length affects reasoning accuracy. Each model has its own optimal prefix length for peak performance: Llama-3.1-8B-Instruct achieves its highest average accuracy (52.0%) with an 8-token prefix, whereas Qwen2.5-Math-7B-Instruct remains stable over prefix lengths from 8 to 32 tokens. The math-specialized Qwen2.5-Math-7B-Instruct is less sensitive to prefix length, likely because it was trained on the high-quality large-scale math corpus, which reinforces its stability on the MATH500 test set. For subsequent experiments, we set Llama-3.1-8B-Instruct to 8 tokens and Qwen2.5-Math-7B-Instruct to 32 tokens. In the following experiments, we set the prefix length to 8 for Llama-3.1-8B-Instruct and 32 for Qwen2.5-Math-7B-Instruct. Because the responses from the long-reasoning DeepSeek-R1-Distill-Qwen-7B are generally more than four times longer than those of Qwen2.5-Math-7B-Instruct, we use a prefix length of 128 tokens for the former.

**Impact of Structure Tuning Ratio**  Figure 3(b) plots the effect of the structure tuning ratio ($p$). The performance of both models generally improves with increasing $p$, peaking at $p = 10\%$. This result supports our hypothesis that even minimal structural supervision effectively guides models towards generating complete responses. However, exceeding this ratio leads to a performance decrease. Consequently, we set $p$ to 0.1 in UPFT for all datasets, except LIMO. For the smaller LIMO dataset, we increased $p$ to 0.3 for structure tuning.

## 4.3 Unsupervised Fine-Tuning

Table 1 presents the results for the unsupervised sampling setting, where the data is not filtered based on the correctness of the extracted answer. We compare these outcomes to those obtained using conventional SFT [21], which fine-tunes models on training data without any self-improvement or test-time verification.

**UPFT demonstrates superior performance compared to SFT in unsupervised fine-tuning.** UPFT consistently outperforms conventional SFT across various self-training datasets and backbone models under the unsupervised sampling setting. For instance, using the U-Hard dataset with Qwen2.5-Math-7B-Instruct, UPFT achieves an average accuracy of 54.5% across all benchmarks, exceeding the 51.3% attained by conventional SFT. Likewise, with DeepSeek-R1-Distill-Qwen-7B and U-Hard, UPFT attains an average of 61.6%, whereas SFT achieves 56.4%. These results highlight UPFT's effectiveness in leveraging unsupervised data to enhance reasoning, thereby reducing the reliance on labeled data. They also indicate the versatility and broad applicability of UPFT across

Table 1: Model performance under the **unsupervised sampling setting**, without any filtering based on the correctness of the final answer. "Length" denotes the average length of the tuning samples in each dataset. Models trained with SFT and the proposed UPFT generate responses of similar length.

| Fine-Tuning | | | Testsets | | | | |
|---|---|---|---|---|---|---|---|
| **Method** | **Data** | **Avg. Length** | **GSM8K** | **MATH500** | **AIME2024** | **GPQA** | **Ave.** |
| *Llama-3.1-8B-Instruct* | | | 82.0 | 51.0 | 3.3 | 8.6 | 36.2 |
| + SFT | PRM | 175.8 | 83.8 | 48.4 | 3.3 | 8.6 | 36.0 |
| + UPFT | (12K) | 15.8 | **85.4** | **52.0** | **6.7** | **9.1** | **38.3** |
| *Qwen2.5-Math-7B-Instruct* | | | 95.2 | 84.0 | 16.7 | 9.6 | 51.4 |
| + SFT | PRM | 300.1 | 95.8 | 83.4 | 13.3 | 9.1 | 50.4 |
| + UPFT | (12K) | 51.4 | 95.5 | 85.6 | 20.0 | 9.6 | **52.6** |
| + SFT | OMI2 | 533.2 | 95.4 | 83.4 | 13.3 | 6.6 | 49.7 |
| + UPFT | (600K) | 67.5 | 95.4 | **86.4** | 20.0 | 9.6 | **52.9** |
| + SFT | LIMO | 491.8 | 95.8 | 84.2 | 20.0 | 7.6 | 51.9 |
| + UPFT | (0.8K) | 77.8 | 95.6 | 85.8 | 20.0 | 8.6 | 52.5 |
| + SFT | U-Hard | 393.3 | 95.5 | 83.4 | 16.7 | 9.6 | 51.3 |
| + UPFT | (100K) | 68.2 | **96.0** | 85.6 | **26.6** | 9.6 | **54.5** |
| *DeepSeek-R1-Distill-Qwen-7B* | | | 88.6 | 87.0 | 40.0 | 13.1 | 57.2 |
| + SFT | LIMO | 2029.5 | 89.7 | 87.0 | 40.0 | 12.1 | 57.2 |
| + UPFT | (0.8K) | 757.7 | **92.0** | **89.4** | 43.3 | **17.7** | 60.6 |
| + SFT | U-Hard | 3440.4 | 89.7 | 87.0 | 36.7 | 12.1 | 56.4 |
| + UPFT | (100K) | 561.7 | 91.4 | 89.2 | **50.0** | 15.7 | **61.6** |

different LLM architectures, supporting its claim of being a flexible, universal method for improving reasoning capabilities.

**The benefits of UPFT are more pronounced on complex reasoning tasks.**   The performance gains of UPFT over conventional SFT are especially evident on more challenging benchmarks such as AIME2024 and GPQA. For example, on AIME2024 with the U-Hard dataset, UPFT achieves 26.6% accuracy with Qwen2.5-Math-7B-Instruct, a notable improvement over the 16.7% achieved by conventional SFT. A similar trend is observed with DeepSeek-R1-Distill-Qwen-7B on AIME2024, where UPFT reaches 50.0% compared with SFT's 36.7%, showing UPFT's capability to improve reasoning on difficult problems, aligning with our claim of enhancing reasoning capabilities effectively and efficiently.

**The U-Hard dataset maximizes UPFT's potential through difficulty-focused curation.**   Training with U-Hard yields the strongest performance improvements across models, particularly for complex benchmarks. Qwen2.5-Math-7B achieves 26.6% on AIME2024 with U-Hard - 10 points higher than with PRM data. This demonstrates that challenging questions provide richer signals for prefix-based self-improvement, as they demand more sophisticated initial reasoning setups. Using U-Hard, UPFT achieves the highest average accuracy with both Qwen2.5-Math-7B-Instruct and DeepSeek-R1-Distill-Qwen-7B, surpassing other datasets such as PRM-12K and OMI2-600K. These findings demonstrate that UPFT effectively leverages diverse and challenging questions to refine the model's reasoning skills in an unsupervised manner, showcasing its data efficiency and versatility.

**UPFT achieves dramatic efficiency gains through minimal token training.**   UPFT reduces training sequence length by 82.6-94.7% compared to SFT across datasets, with U-Hard samples averaging 68.2 tokens vs. SFT's 393.3. This directly translates to 6.3-16.7x faster training iterations and reduced memory consumption. Notably, DeepSeek-R1-Distill-Qwen-7B attains better performance with UPFT's 561-token samples than SFT's 3,440-token sequences, proving that strategic prefix selection preserves critical learning signals. These efficiency characteristics validate UPFT's practical value for resource-constrained applications while maintaining or exceeding baseline performance.

Table 2: Model performance under the **supervised sampling setting** on the PRM-12K dataset, with filtering 16 sampled solutions based on the correct extract answer. "#Tokens" denote the number of tokens in each phase. Compared to RFT, V-STaR requires two additional samples to tune the verifier.

| Method | #Tokens | | Testsets | | | | |
|---|---|---|---|---|---|---|---|
| | Sampling | Tuning | GSM8K | MATH500 | AIME2024 | GPQA | Ave. |
| *Llama-3.1-8B-Instruct* | | | 82.0 | 51.0 | 3.3 | 8.6 | 36.2 |
| + RFT | 36.9M | 2.3M | 86.0 | 52.0 | 6.7 | 9.1 | 38.5 |
| + V-STaR | | 6.8M | 85.4 | 52.6 | 6.7 | 8.6 | 38.3 |
| + UPFT (*Ours*) | 0.2M | | 85.4 | 52.0 | 6.7 | 9.1 | 38.3 |
| + Lable Filter | 36.9M | 0.2M | 85.8 | 53.4 | 6.7 | 9.1 | **38.8** |
| *Qwen2.5-Math-7B-Instruct* | | | 95.2 | 84.0 | 16.7 | 9.6 | 51.4 |
| + RFT | 51.7M | 3.2M | 95.7 | 85.2 | 20.0 | 9.6 | 52.6 |
| + V-STaR | | 9.6M | 96.0 | 85.4 | 20.0 | 10.1 | **52.9** |
| + UPFT (*Ours*) | 0.6M | | 95.5 | 85.6 | 20.0 | 9.6 | 52.6 |
| + Lable Filter | 51.7M | 0.6M | 96.0 | 85.6 | 20.0 | 10.1 | **52.9** |
| *DeepSeek-R1-Distill-Qwen-7B* | | | 88.6 | 87.0 | 40.0 | 13.1 | 57.2 |
| + RFT | 318.0M | 19.9M | 90.7 | 87.0 | 40.0 | 11.1 | 57.2 |
| + UPFT (*Ours*) | 5.0M | | 91.9 | 88.4 | 40.0 | 14.6 | 58.7 |
| + Lable Filter | 318.0M | 4.5M | 92.3 | 89.2 | 40.0 | 13.6 | **58.8** |

## 4.4 Supervised Fine-Tuning

In the supervised sampling setting, we compare our approach with two SFT variants in Table 2:

- RFT [33] samples 16 candidate responses with a label filter to identify a correct one.

- V-STaR [11] produces 16 responses for each tuning instance, trains a verifier on them for one iteration, and uses it at test time to select the answer from 4 completions.

**UPFT achieves competitive performance with supervised methods while requiring significantly fewer tokens in both sampling and tuning.** Across multiple model architectures, UPFT matches or exceeds the performance of supervised baselines such as RFT and V-STaR. On the Llama-3.1-8B-Instruct backbone, UPFT attains an average reasoning benchmark score of 38.3%, matching V-STaR (38.3%) and approaching RFT (38.5%). For Qwen2.5-Math-7B-Instruct, UPFT achieves identical average accuracy to RFT (52.6%), while using only 1.2% of the sampling tokens (0.6M vs. 51.7M). Most notably, UPFT enables DeepSeek-R1 to achieve 58.7% average accuracy, which is 1.5 points higher than RFT, while requiring 16× fewer tuning tokens than RFT. This underscores how prefix-based fine-tuning effectively captures critical reasoning patterns without incurring the computational overhead of sampling 16 responses per question, as required by supervised baselines. Moreover, UPFT maintains its strong performance even when ground-truth answers are available. By focusing on the shared reasoning prefixes, it does not sacrifice accuracy. These results reinforce our core claim: prefix-based training captures the essential reasoning signals available in full trajectories, validating our Prefix Self-Consistency hypothesis. This hypothesis posits that the essential signals for effective reasoning are largely contained within the initial prefixes of successful solution trajectories, allowing for efficient learning and high performance.

**UPFT offers seamless integration with label verification for enhanced accuracy.** An additional benefit of UPFT is its inherent flexibility and compatibility with label verification techniques when such information is accessible. As demonstrated in the table's last rows for each model backbone, when UPFT is enhanced with label filtering, it attains 58.8% average accuracy on DeepSeek-7B, surpassing all baselines while maintaining a 4.4× tuning token advantage over RFT. For Qwen2.5-Math-7B-Instruct, we match V-STaR's 52.9% peak performance without requiring compute-intensive verifier training or best-of-N inference. This dual capability demonstrates UPFT's unique adaptability - it functions as a standalone unsupervised learner while seamlessly integrating supervision when available, making it practical for real-world deployment across the supervision spectrum.

# 5 Related Work

**Unsupervised Learning**   Unsupervised learning encompasses diverse methodologies, including pseudo-labeling [30], pivot-based approaches [22], and adversarial networks [5]. Self-supervised learning has since emerged as a dominant paradigm, particularly for language models, leading to the success of pre-trained models like BERT [14] and others [9, 23]. More recently, self-improvement techniques, such as self-rewarding [2, 32], further advance unsupervised learning by enabling models to iteratively enhance performance without external supervision. Building upon self-improvement, AL [13] extends this paradigm to unsupervised domain adaptation.

**Self-Improvement**   A family of methods, starting with STaR [34], reinforced self-training [7], and rejection fine-tuning [33], leverages the solutions generated by large language models (LLMs) to iteratively update and improve the models themselves. These techniques involve fine-tuning the model on the generated solutions that produce correct answers. ReST$^{EM}$ [25] views this fine-tuning process as expectation-maximization-based reinforcement learning (RL) for a solution-generating agent. Wang et al. [28] propose using a contrastive loss to enhance the likelihood of correct solutions over incorrect ones. The discovery of successful solutions presents a significant exploration challenge. Luong et al. [17] demonstrated that RL-based fine-tuning of an LLM is particularly difficult unless preceded by several steps of supervised fine-tuning. In An et al. [1], a more powerful LLM was employed to edit the incorrect rationales generated by a smaller model, thereby providing positive data for its fine-tuning. However, Huang et al. [12] argued that LLMs have limited capacity to correct their own reasoning flaws.

In contrast to the above approaches, this paper focuses on discovering self-supervised signals specifically for mathematical reasoning. We introduce UPFT, a simple and effective unsupervised post-training method requiring only questions and the LLM itself.

# 6 Conclusion

In this work, we presented an unsupervised fine-tuning method that enhances the reasoning capabilities of large language models using only prefix substrings as minimal guidance. Our approach leverages the inherent reasoning structures within pretrained models, exploiting the phenomenon of Prefix Self-Consistency where different reasoning trajectories share common prefixes. Extensive experiments demonstrated that our method outperforms traditional full-token fine-tuning and achieves performance comparable to supervised approaches like RFT, with significantly reduced training and inference times. This work highlights the potential of minimal unsupervised fine-tuning in improving the reasoning abilities of LLMs without relying on external supervision or extensive computational resources. Future work will explore the application of this method to other challenging tasks and investigate the theoretical underpinnings of Prefix Self-Consistency in more depth.

## Acknowledgement

This work was supported by Major Frontier Exploration Program (Grant No. C10120250085) from the Shenzhen Medical Academy of Research and Translation (SMART), the Shenzhen Science and Technology Program (JCYJ20220818103001002), NSFC grant 72495131, Shenzhen Doctoral Startup Funding (RCBS20221008093330065), Tianyuan Fund for Mathematics of National Natural Science Foundation of China (NSFC) (12326608), Shenzhen Science and Technology Program (Shenzhen Key Laboratory Grant No. ZDSYS20230626091302006), and Shenzhen Stability Science Program 2023.

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

# A  Settings of the Experiments

## A.1  Task Template for Prefix Tuning

For the prefix fine-tuning, we simply apply the task template shown in Figure 4.

> ***Task template used for prefix tuning.***
>
> `[question]` Please provide the initial step towards resolving the question. This step may serve as a foundation but might not encompass the entire solution.

Figure 4: The task template used to learn from the prefix of the reasoning traces. `[question]` represents the question that needs to be answered.

# B  Hyperparameters

Table 3: The hyperparameters used for our method on all training corpora.

| Hyperparam. | PRM-12K | OMI2-60K | LIMO | U-Hard |
|---|---|---|---|---|
| Optimizer | | AdamW | | |
| Warmup Ratio | | 0.03 | | |
| Learning Rate | | 1e-6 | 2e-6 | 1e-6 |
| LR Schedule | | constant_with_warmup | | |
| Batch Size | | 1 | | |
| Gradient Step | | 8 | | |
| Max Length | | 4096 | 16384 | |
| # Epoch | 2 | 1 | 3 | 1 |

To ensure a fair comparison, UPFT employs the same hyperparameters as conventional SFT when fine-tuning models, as listed in Table 3. During the inference stage, we adopt a prompted zero-shot setup and use standard greedy decoding, wherein models are directed to answer each question using natural language instructions without any accompanying contextual demonstrations.

# C  Limitations

- **Validation with larger-scale models:** In our experiments, the choice of backbone was primarily based on the mathematical capabilities of the model. Our goal is to further enhance the model's reasoning ability at a low cost. In future research within the RL framework, we will consider validating our approach on larger-scale models, such as QwQ-32B-Preview and DeepSeek-R1-Distill-Qwen-32B.

- **Study on sample-independent prefix selection strategy:** Despite achieving satisfactory performance, this work is based on a heuristic prefix selection strategy. In future research, we will continue to design an adaptive sample-independent prefix selection for UPFT.

# D  Case study of Prefix Data

We provide a case study of prefix data shown in Figure 5. We can see that the different 16 rollout samples have very similar prefixes.



**Case Study**

**Question:** Melinda has three empty boxes and $12$ textbooks, three of which are mathematics textbooks. One box will hold any three of her textbooks, one will hold any four of her textbooks, and one will hold any five of her textbooks. If Melinda packs her textbooks into these boxes in random order, the probability that all three mathematics textbooks end up in the same box can be written as $\frac{m}{n}$ , where $m$ and $n$ are relatively prime positive integers. Find $m + n$.

**The first 32 words (prefix substring) of all answers:**

A1:  To determine the probability that all three mathematics textbooks end up in the same box, we need to consider the total number of ways to distribute the 12 textbooks into the three

A2:  To determine the probability that all three mathematics textbooks end up in the same box, we need to follow these steps:  1.  **Calculate the total number of ways to pack the textbooks

A3:  To determine the probability that all three mathematics textbooks end up in the same box, we start by calculating the total number of ways to distribute the 12 textbooks into the three

A4:  To determine the probability that all three mathematics textbooks end up in the same box, we need to consider the total number of ways to distribute the 12 textbooks into the three

A5:  To determine the probability that all three mathematics textbooks end up in the same box, we start by calculating the total number of ways to distribute the 12 textbooks into the three

A6:  To determine the probability that all three mathematics textbooks end up in the same box, we need to consider the total number of ways to distribute the 12 textbooks into the three

A7:  To determine the probability that all three mathematics textbooks end up in the same box, we need to consider the total number of ways to distribute the 12 textbooks into the three

A8:  To determine the probability that all three mathematics textbooks end up in the same box, we need to consider the total number of ways to distribute the 12 textbooks into the three

A9:  To determine the probability that all three mathematics textbooks end up in the same box, we need to consider the total number of ways to distribute the 12 textbooks into the three

A10:  To determine the probability that all three mathematics textbooks end up in the same box, we need to consider the total number of ways to distribute the 12 textbooks into the three

A11:  To determine the probability that all three mathematics textbooks end up in the same box, we start by calculating the total number of ways to distribute the 12 textbooks into the three

A12:  To determine the probability that all three mathematics textbooks end up in the same box, we need to follow these steps:  1.  **Calculate the total number of ways to distribute the 12

A13:  To determine the probability that all three mathematics textbooks end up in the same box, we need to consider the total number of ways to distribute the 12 textbooks into the three

A14:  To determine the probability that all three mathematics textbooks end up in the same box, we need to follow these steps:  1.  **Calculate the total number of ways to pack the textbooks

A15:  To determine the probability that all three mathematics textbooks end up in the same box, we need to consider the total number of ways to distribute the 12 textbooks into the three

A16:  To determine the probability that all three mathematics textbooks end up in the same box, we need to consider the total number of ways to distribute the 12 textbooks into the three



Figure 5: With the temperature set to 0.7, we sample 16 times based on Qwen2.5-Math-7B-Instruct for the given question, where A1-A16 represents the corresponding output results.

