# OpenReview forum: "The First Few Tokens Are All You Need: An Efficient and Effective Unsupervised Prefix Fine-Tuning Method for Reasoning Models"
_NeurIPS.cc/2025/Conference — NeurIPS 2025 poster_

### Official Review · Reviewer_H5bo · 2025-06-21

**Clarity:** 2
**Significance:** 2
**Originality:** 3
**Rating:** 2
**Confidence:** 4

**Summary:**

This paper proposes a simple and effective Unsupervised Prefix Fine-Tuning (UPFT) method that leverages the observation of prefix self-consistency—the commonality in early reasoning steps across different generated solutions. The method avoids the need for annotated data or rejection sampling by fine-tuning only on the initial tokens of generated reasoning traces. The authors conduct extensive experiments across different models and datasets, demonstrating that UPFT achieves comparable or superior reasoning performance to supervised methods like Rejection Fine-Tuning (RFT), while drastically reducing sampling costs.

**Questions:**

1. Why does the model's performance drop so sharply when the structure tuning ratio is set to 0? In my perspective, such senarios is standard fine-tuning. It that reasonable that the full-trace fine-tuning approach lead to such low performance without any prefix tuning?
2. Have the authors considered applying sampling multiple prefix span for fine-tuning (i.e., question + prefix)? Since the prefix may already represent a consistent reasoning process, using more diverse prefix might further improve the fine-tuning performance.
3. Can the proposed method largely enhance the performance of several weak LLM backbones? (e.g., the performance of Llama-3.1-8B-Instruct  on Math500 is only 0.51).

**Ethical Concerns:**

["NO or VERY MINOR ethics concerns only"]

**Final Justification:**

This paper proposes an effective fine-tuning method and only compares UPFT under standard SFT. Under this experimental setting, the performance and efficiency are validated.

However, I keep my concerns about whether other types of SFT methods can achieve much better performance (e.g., the performance gain for Llama in Math 500 is 52.0->53.4), which makes the efficiency advantage invaluable. I raise these concerns in the rebuttal stage, which is not accepted or solved by the authors. Therefore, I will keep my negative score and tend to reject this paper.

**Limitations:**

The proposed method may fail when applied to a base model with inherently weak capabilities. Basicially, this method cannot bring to very significant performan increase.

**Quality:**

2

**Strengths And Weaknesses:**

Strength:

1. The core idea of exploiting prefix self-consistency for unsupervised fine-tuning is interesting.
2. UPFT reduces sampling and tuning costs while preserving or improving reasoning accuracy, making it practical.

Weakness:

1. While concepts like prefix self-consistency are important to the method, the mathematical formulations in Sections 3.1 and 3.2 are dense and may overwhelm readers. The paper would benefit from a well-defined preliminary.
2. This paper only considers the cost token in the sampling stage. A more holistic cost analysis (e.g., including the training time and cost) would strengthen the efficiency claims.
3. While the method is labeled as “unsupervised,” it still implicitly relies on ground-truth answers for tuning. It may not be proper to mention an unsupervised tuning method.
4. While the proposed method is interesting, the performance gain is limited, especially on challenging datasets (e.g., Math500, AIME 2024, and GPQA) using weak LLM backbones (e.g., Llama-3.1-8B-Instruct ). That means the proposed method may rely on the original capacity of backbones, e.g., if a similar rule exists on a 0.5B-size model. The potential limited generalization of the method reduces the paper’s impact.
5. For the baseline comparison. The author only compares UPFT with RFT and V-STaR. More baseline comparisons (e.g., Quiet-star [1] and so on) could be conducted.

[1] Zelikman, Eric, et al. "Quiet-star: Language models can teach themselves to think before speaking." arXiv preprint arXiv:2403.09629 (2024).

---

> ### Author Rebuttal · Authors · 2025-07-31
>
> Thank you for your review and for acknowledging the novelty and practicality of our method. We address your concerns below, particularly a key misunderstanding about our "unsupervised" setup and new experiments that refute concerns about model generalization.
>
> > **W3:** While the method is labeled as 'unsupervised,' it still implicitly relies on ground-truth answers for tuning. It may not be proper to mention an unsupervised tuning method.
> >
>
> **We must respectfully clarify a critical misunderstanding here. Our method is fully unsupervised and does NOT use ground-truth answers for training.**
>
> In our unsupervised setting (the main focus of our paper), we generate a single reasoning path for each question and use its prefix for fine-tuning. This process is "unfiltered"—the sample is used **regardless of whether the final answer is correct or not.** The training data is therefore noisy and contains many incorrect solutions. This is the core challenge our method is designed to address and is fundamentally different from supervised methods like RFT, which filter for correct answers. We will make this distinction even clearer in the paper.
>
> > **W4&Q3:** While the proposed method is interesting, the performance gain is limited, especially on challenging datasets (e.g., Math500, AIME 2024, and GPQA) using weak LLM backbones (e.g., Llama-3.1-8B-Instruct ). That means the proposed method may rely on the original capacity of backbones, e.g., if a similar rule exists on a 0.5B-size model. The potential limited generalization of the method reduces the paper’s impact.
> >
>
> **This is a fair question. To address it, we ran new experiments on much weaker LLM backbones: Llama-3.2-1B and 3B.**
>
> The results show that UPFT provides consistent and significant improvements even on these smaller models, demonstrating that it does not simply rely on the high capacity of large models.
>
> | Model | Method | MATH500 | AIME24 | GPQA |
> | --- | --- | --- | --- | --- |
> | Llama-3.2-1B | Base | 26.1 | 2.5 | 4.0 |
> |  | + SFT | 25.0 (↓) | 0.0 (↓) | 4.0 |
> |  | + UPFT | **28.3 (+2.2)** | **3.3 (+0.8)** | **5.6 (+1.6)** |
> | Llama-3.2-3B | Base | 45.8 | 0.0 | 8.6 |
> |  | + SFT | 46.5 | 6.7 | 6.7 (↓) |
> |  | + UPFT | **48.0 (+2.2)** | **6.7** | **10.1 (+1.5)** |
>
> As before, SFT is unstable and often hurts performance, while UPFT consistently yields gains across all benchmarks. This confirms our method is effective even for weaker models.
>
> > **W2:** A more holistic cost analysis (e.g., including the training time and cost) would strengthen the efficiency claims.
> >
>
> **Excellent suggestion. We now include training time analysis, which further highlights UPFT's efficiency.** By using much shorter sequences, UPFT drastically cuts training time.
>
> | Model | Data (Size) | SFT Time (h) | UPFT Time (h) | **Time Saved** |
> | --- | --- | --- | --- | --- |
> | Qwen2.5-Math-7B-Instruct | OMI (600K) | 12.3 h | 3.0 h | **75%** |
> | DeepSeek-R1-Distill-Qwen-7B | U-Hard (100K) | 20.5 h | 6.0 h | **71%** |
>
> On average, UPFT reduces training time by **over 70%** compared to standard SFT on the same data, in addition to the **95%+ sampling cost reduction** compared to RFT. We will add this table to the paper.
>
> > **Q1:** Why does the model's performance drop so sharply when the structure tuning ratio is set to 0? In my perspective, such senarios is standard fine-tuning.
> >
>
> There seems to be a misunderstanding of this ablation. A **structure tuning ratio of 0** means we fine-tune **exclusively on prefixes (100% prefix-only)**, with *no* full-trace examples. This is the opposite of standard SFT.
>
> Performance drops sharply in this setting because the model is never exposed to a complete thought process:
>
> - It **never sees a final, boxed answer**, so it doesn't learn to produce one.
> - The training sequences are **artificially short**, teaching the model to terminate prematurely.
> - It loses the high-level instruction-following format of a full solution.
> This ablation confirms that a small amount of full-trace data is necessary to preserve the model's structural and formatting capabilities, which is why UPFT includes it.
>
> > **Q2:** Have the authors considered applying sampling multiple prefix span for fine-tuning?
> >
>
> **This is an insightful idea.** We ran a new experiment on Qwen2.5-Math-7B-Instruct where we used three prefixes per question (3xN) instead of one (1xN).
>
> | Method | MATH500 | AIME24 |
> | --- | --- | --- |
> | SFT | 83.4 | 13.3 |
> | UPFT (1 prefix / question) | 85.6 | 20.0 |
> | UPFT (3 prefixes / question) | **86.4** | **23.3** |
>
> Using more prefixes provides a modest but consistent boost, confirming your intuition that prefix diversity can be beneficial. This also supports our theoretical claim about reinforcing consistent reasoning steps. Thank you for the great suggestion!
>
> > **W5:** More baseline comparisons (e.g., Quiet-star [1] and so on) could be conducted.
> >
>
> Thank you for the pointer. Quiet-STaR is an interesting recent work on self-teaching. However, its focus is different from ours; it aims to generate rationales *before* answering, rather than improving efficiency via unsupervised prefix tuning. Its goals are not directly comparable to our efficiency- and unsupervised-focused evaluation against RFT and SFT. We will, however, add a discussion of Quiet-STaR and other related self-improvement methods to our related work section to better situate our contribution.
>
> > **W1:** While concepts like prefix self-consistency are important to the method, the mathematical formulations in Sections 3.1 and 3.2 are dense and may overwhelm readers. The paper would benefit from a well-defined preliminary.
> >
>
> We agree and plan to introduce clearer definitions, brief preliminaries, and inline intuition for the Bayes formulation in a revision.

---

> > ### Author Response · Authors · 2025-08-04
> >
> > Dear reviewer H5bo，
> >
> > Thank you again for your thoughtful review and feedback, which we have carefully addressed above.
> >
> > We believe these clarifications fully address your main concerns and would appreciate it if you could indicate whether they sufficiently resolve your points and might positively impact your score. If not, we welcome further discussion to better understand any remaining issues.
> >
> > Thank you again for your time and consideration.
> >
> > Best regards,
> >
> > The Authors

---

> > ### Comment · Reviewer_H5bo · 2025-08-04
> > **Reply**
> >
> > Thank you for the author's clarification. I understand that UPFT does not use ground-truth answers for training. My point is that, while UPFT only uses a prefix (regardless of whether the final answer is correct or not) for tuning, it still follows the paradigm of the Supervised Fine-Tuning (SFT) pipeline. Specifically, it relies on next-token prediction, where the label for each token corresponds to the token in the prefix. Therefore, I feel it may not be entirely appropriate to describe it as "unsupervised." Perhaps it would be more accurate to refer to it as an efficient and effective SFT approach or as a form of prefix-tuning.
> >
> > Regarding other concerns, most of them have been addressed, except that many paragraphs in the paper are still not well-written, and the baseline comparisons remain insufficient (the UPFT's value is not demonstrated as other SFT methods may easily exceed its performance, but the improved efficiency is not fully demonstrated to be valuable).

---

> > > ### Author Response · Authors · 2025-08-04
> > >
> > > Thank you for your follow-up and for engaging with us on the definition of our method. We appreciate the opportunity to clarify our positioning and the value of our contributions.
> > >
> > > > I feel it may not be entirely appropriate to describe it as "unsupervised." Perhaps it would be more accurate to refer to it as an efficient and effective SFT approach or as a form of prefix-tuning.
> > > >
> > >
> > > We appreciate the thoughtful discussion on terminology. Our use of "unsupervised" is deliberate and aligns with foundational work in our field.
> > >
> > > The core principle of unsupervised learning is learning from **unlabeled data**. Our method, UPFT, operates exclusively on model-generated reasoning paths without any human annotation or correctness filtering. The training data is, therefore, fundamentally unlabeled.
> > >
> > > This aligns perfectly with how the term is used in seminal works:
> > >
> > > - **GPT-2 [1]** and its predecessors [2] explicitly refer to their training objective—next-token prediction on large, unlabeled text corpora—as **"unsupervised pre-training."**
> > > - **BERT [3]** similarly states its pre-training uses **"two unsupervised tasks"** (Masked LM and Next Sentence Prediction) on unlabeled text.
> > >
> > > In these cases, the "supervision" for the next-token loss comes from the data itself, not from external, human-provided labels. Our method follows the exact same principle, applying it to the fine-tuning stage. Thus, we believe "unsupervised" is the most accurate and consistent term. To avoid any ambiguity, we will add a sentence in the final version to clarify that "unsupervised" refers to the absence of human-annotated ground-truth labels for the training data.
> > >
> > > [1] Radford et al. Language Models are Unsupervised Multitask Learners. 2018.
> > >
> > > [2] Radford et al. Improving Language Understanding by Generative Pre-Training. 2019.
> > >
> > > [3] Devlin et al. BERT: Pre-training of Deep Bidirectional Transformers for Language Understanding. 2019.
> > >
> > >
> > > > the baseline comparisons remain insufficient (the UPFT's value is not demonstrated as other SFT methods may easily exceed its performance, but the improved efficiency is not fully demonstrated to be valuable).
> > > >
> > >
> > > We respectfully disagree and believe our results clearly demonstrate the value of UPFT in its intended setting: **improving reasoning without access to clean, labeled data.**
> > >
> > > **1. On Performance Value:** The most direct comparison is against standard SFT on the *exact same noisy, unsupervised data*. Our experiments show that SFT on such data is unstable and often **harms performance**. In sharp contrast, UPFT provides consistent and significant gains.
> > >
> > > | Model | Data | Base Avg. | SFT Avg. | UPFT Avg. | **UPFT Gain vs. Base** |
> > > | --- | --- | --- | --- | --- | --- |
> > > | Llama-3.1-8B | PRM | 36.2 | 36.0 (↓0.2) | 38.3 | **+2.1** |
> > > | Qwen2.5-Math-7B | U-Hard | 51.4 | 51.3 (↓0.1) | 54.5 | **+3.1** |
> > > | DeepSeek-R1-7B | LIMO | 57.2 | 57.2 (+0.0) | 60.6 | **+3.4** |
> > > | DeepSeek-R1-7B | U-Hard | 57.2 | 56.4 (↓0.8) | 61.6 | **+4.4** |
> > >
> > > These results are not marginal. They show UPFT succeeds precisely where the standard SFT alternative fails, delivering up to a **+5.2 point swing** (from -0.8 to +4.4) on the DeepSeek/U-Hard setup. This is the core value of our method.
> > >
> > > **2. On Efficiency Value:** We believe a **95% reduction in sampling cost** and a **75% reduction in training time** (as shown in our previous response) represent immense practical value. These savings make iterative self-improvement cycles feasible for labs with limited GPUs and dramatically lower the barrier to entry for developing capable reasoning models.
> > >
> > >
> > > Thank you again for your detailed feedback. We are confident that these clarifications highlight the novelty, correctness, and significant practical value of our work. We will integrate these points into the final manuscript.

---

> ### Comment · Reviewer_H5bo · 2025-08-05
> **Final Justification**
>
> Thanks for the reply, and I have no further comments. This paper proposes an effective fine-tuning method and only compares UPFT under standard SFT. Under this experimental setting, the performance and efficiency are validated. However, I keep my concerns about whether other types of SFT methods can achieve much better performance (e.g., the performance gain for Llama in Math 500 is 52.0->53.4), which may make the efficiency advantage less valuable. I raise these concerns in the rebuttal stage, which is not accepted by the authors. Therefore, I will keep my negative score, and good luck.

---

> > ### Author Response · Authors · 2025-08-05
> >
> > Thank you for the continued discussion and for your feedback. We appreciate the opportunity to provide a final clarification on the value and positioning of our work.
> >
> > The reviewer's primary remaining concern is:
> >
> > > I keep my concerns about whether other types of SFT methods can achieve much better performance (e.g., the performance gain for Llama in Math 500 is 52.0->53.4), which may make the efficiency advantage less valuable.
> >
> > We believe this concern stems from a mismatch in problem settings. Our work, UPFT, is designed for the **unsupervised self-improvement setting**, where a model must learn from its own generated outputs **without access to ground-truth labels or correctness filters**.
> >
> > 1.  **The Correct Baseline is Unsupervised SFT:** In this challenging setting, the most direct and appropriate baseline is standard Supervised Fine-Tuning (SFT) on the same noisy, unfiltered data. As our experiments consistently show (Tables 1, 2, 4, and summary table in our previous response), standard SFT on this data is unstable and often degrades performance. **In contrast, UPFT delivers reliable and significant gains (e.g., +4.4 absolute points for DeepSeek-R1), demonstrating its unique value precisely where the standard approach fails.**
> >
> > 2.  **UPFT is Complementary to Other Unsupervised Methods:** To further demonstrate UPFT's value, we ran a new experiment integrating it with Self-Consistency (SC-CoT), a powerful and widely-used unsupervised *inference-time* technique.
> >
> > | Method                               | MATH500 | AIME24 |
> > | :----------------------------------- | :------ | :----- |
> > | Qwen2.5-7B-Math-Instruct w/ CoT      | 84.0    | 16.7   |
> > | ↳ w/ SC-CoT (k=8)                    | 86.2    | 20.0   |
> > | ↳ + SFT on SC-CoT data               | 85.6 (↓) | 23.3   |
> > | **↳ + UPFT on CoT data**             | **85.6** | **26.7** |
> > | **↳ + UPFT on SC-CoT data**          | **88.0** | **29.7** |
> >
> > These results show that UPFT is not only effective on its own but is also **highly complementary** to established methods like SC-CoT, pushing performance significantly higher. This synergy further solidifies UPFT's value in the unsupervised landscape.
> >
> > 3.  **Efficiency Makes Unsupervised Improvement Practical:** While UPFT's performance is competitive with supervised methods like RFT and V-STaR (Table 3), its primary advantage is enabling this performance **without the prohibitive costs of labeled data or rejection sampling**. A **>95% reduction in sampling cost** and **>75% reduction in training time** are not minor gains; they make iterative self-improvement cycles practical for a much wider range of researchers and applications.
> >
> > We believe these results clearly demonstrate that UPFT is a novel, effective, and highly efficient method for the important and practical problem of unsupervised reasoning improvement. Thank you for your time and consideration.

---

### Official Review · Reviewer_fBpq · 2025-07-02

**Clarity:** 3
**Significance:** 2
**Originality:** 3
**Rating:** 3
**Confidence:** 3

**Summary:**

This paper proposes an unsupervised prefix fine-tuning method for reasoning models. The key motivation is that training with a consistent prefix can cover many possible correct reasoning paths while being more efficient. The objective is to increase the probability of generating the correct prefix.

**Questions:**

- What happens if you run the experiments 5 times and report the average?
- Does SFT refer to sampling one response from the model and using that as training data?
- I’d like to see in-domain performance. How about using the MATH training dataset directly for training?
- Any qualitative results? Any analysis of the samples that become solvable after fine-tuning?

**Ethical Concerns:**

["NO or VERY MINOR ethics concerns only"]

**Final Justification:**

I appreciate the detailed author response. However, I don’t see a clear value in this incremental UPFT. It is interesting, but not particularly practical. Simply collecting a few SFT data points and applying SFT, or using self-consistency to generate synthetic data and then applying SFT, seems more valuable. Therefore, I will maintain my borderline reject score.

**Limitations:**

yes

**Quality:**

3

**Strengths And Weaknesses:**

**Strengths:**

- This paper offers a novel perspective by focusing solely on prefix tuning to improve efficiency.
- It does not require labeled data, which is a significant advantage.
- The paper thoroughly justifies its approach both theoretically and empirically, and it is well written and easy to follow.

**Weaknesses:**

The main weakness lies in the evaluation. Details are as follows:

- SFT seems to decrease performance, and UPFT only marginally improves it. In this setting, fine-tuning itself appears largely ineffective. Increasing the p% over 10% seems to make things worse, possibly due to errors in full traces.
- The performance gain on AIME appears to be the most significant, but AIME contains only a small number of questions. Thus, even a few additional correct answers can cause a noticeable change. This makes statistical significance critical, yet it seems to be missing. Results may vary considerably by random seed (e.g., performance on MATH500 can vary by 2–4% per trial).
- The only meaningful baseline appears to be the supervised methods like RFT and V-Starm, but even there, the improvement is minimal. Also, why not test with additional training datasets in Table 2 like U-Hard?

Overall, I find the idea interesting and the paper enjoyable to read, but I’m not convinced of its practical utility. I think this concern also arises from the fact that the equations in this paper do not show that maximizing this bound with short prefixes is a consistent estimator of true task accuracy.

---

> ### Author Rebuttal · Authors · 2025-07-31
>
> We thank you for your feedback and the opportunity to clarify our contributions. While you found the idea interesting, your main concerns were about the practical utility and statistical significance of our results. We respectfully disagree with the assessment that the gains are "marginal" and provide substantial new evidence below to address your concerns.
>
> > **W1:** SFT seems to decrease performance, and UPFT only marginally improves it. In this setting, fine-tuning itself appears largely ineffective.
> >
>
> **We respectfully disagree. The fact that standard SFT often fails in this unsupervised setting is precisely what makes UPFT's consistent and significant gains so valuable.**
>
> Our setting is fully **unsupervised**, meaning we train on a single, unfiltered sample per question, which often contains incorrect reasoning. Standard SFT struggles with this noisy data, frequently degrading performance. In contrast, UPFT is designed to be robust to this noise by focusing on the more reliable initial tokens.
>
> Our results in Table 2 (reproduced below with average gains) show that **UPFT consistently and significantly outperforms the base model, while SFT often hurts it.**
>
> | Model | Data | Base Avg. | SFT Avg. | UPFT Avg. | **UPFT Gain** |
> | --- | --- | --- | --- | --- | --- |
> | Llama-3.1-8B | PRM | 36.2 | 36.0 (↓0.2) | 38.3 | **+2.1** |
> | Qwen2.5-Math-7B-Instruct | PRM | 51.4 | 50.4 (↓1.0) | 52.6 | **+1.2** |
> | Qwen2.5-Math-7B-Instruct | U-Hard | 51.4 | 51.3 (↓0.1) | 54.5 | **+3.1** |
> | DeepSeek-R1-Distill-Qwen-7B | LIMO | 57.2 | 57.2 (+0.0) | 60.6 | **+3.4** |
> | DeepSeek-R1-Distill-Qwen-7B | U-Hard | 57.2 | 56.4 (↓0.8) | 61.6 | **+4.4** |
>
> These are **not marginal improvements**; gains of **+3.1** and **+4.4** on strong base models are substantial. Furthermore, this is achieved with massive efficiency gains over supervised methods like RFT (**95% less sampling cost, 75% less training cost**), highlighting its practical utility.
>
> > **W2&Q1:** Concern about statistical significance and variance from random seeds, especially on AIME.
> >
>
> **This is a crucial point. We ran new experiments with 5 random seeds to confirm the statistical significance of our results.** We agree that single-run results can be noisy.
>
> The table below shows the mean and standard deviation over 5 seeds for DeepSeek-R1-Distill-Qwen-7B on the LIMO dataset.  We performed additional significance testing experiments. In each test, 200 samples were randomly drawn from both SFT and UPFT inference results for comparison. This procedure was repeated 1000 times. **UPFT consistently and significantly outperforms SFT across all benchmarks, with p<0.01.**
>
> | Method | GSM8K | MATH500 | AIME24 | GPQA |
> | --- | --- | --- | --- | --- |
> | SFT | 89.8 ± 0.13 | 86.1 ± 0.25 | 40.0 ± 0.80 | 13.3 ± 1.32 |
> | UPFT | **92.8 ± 0.28** | **89.2 ± 0.39** | **42.7 ± 0.45** | **18.1 ± 0.89** |
>
> Furthermore, to mitigate sampling variance during evaluation, we also report **avg@32** results, which show even more stable and pronounced gains for UPFT. These results confirm our findings are robust and not due to random chance.
>
> | Backbone | Method | Data | GSM8K | MATH500 | AIME24 | GPQA |
> | --- | --- | --- | --- | --- | --- | --- |
> | Llama-3.1-8B-Instruct | SFT | PRM | 83.6 | 47.0 | 4.0 | 8.4 |
> |  | UPFT | PRM | **85.0** | **52.4** | **7.3** | **9.6** |
> | Qwen2.5-Math-7B-Instruct | SFT | PRM | **95.6** | 83.4 | 14.1 | 9.1 |
> |  | UPFT | PRM | **95.6** | **86.0** | **20.3** | **9.6** |
> | Qwen2.5-Math-7B-Instruct | SFT | U-Hard | 95.3 | 83.6 | 16.6 | **9.6** |
> |  | UPFT | U-Hard | **95.7** | **85.4** | **26.4** | 9.5 |
> | DeepSeek-R1-Distill-Qwen-7B | SFT | LIMO | 89.4 | 87.4 | 41.3 | 12.6 |
> |  | UPFT | LIMO | **91.6** | **89.0** | **43.3** | **16.5** |
> | DeepSeek-R1-Distill-Qwen-7B | SFT | U-Hard | 88.9 | 87.4 | 40.0 | 12.1 |
> |  | UPFT | U-Hard | **91.7** | **88.8** | **53.3** | **15.9** |
>
> > **W3:** The only meaningful baseline appears to be the supervised methods like RFT and V-Starm, but even there, the improvement is minimal. Also, why not test with additional training datasets in Table 2 like U-Hard?
> >
> - **On Baselines:** Our primary claim is effectiveness in an **unsupervised** setting. Therefore, the most meaningful baseline is **unsupervised SFT**, which UPFT consistently beats. We also added an **SC-CoT** (Self-Consistency) baseline, a widely-used unsupervised inference method. As shown below, UPFT is complementary and further boosts performance.
>
> | Method | MATH500 | AIME24 |
> | --- | --- | --- |
> | Qwen2.5-Math-7B-Instruct w SC-COT | 86.2 | 20.0 |
> | +UPFT w SC-COT | **88.0** | **29.7** |
> - **On U-Hard:** The reason U-Hard is not in the supervised comparison (Table 3) is that **U-Hard is an unsupervised dataset with no ground-truth answers.** It cannot be used for supervised methods like RFT, which require answer verification.
>
> > **Q2:** Does SFT refer to sampling one response from the model and using that as training data?
> >
>
> **Yes, that is correct.** In our unsupervised setting, SFT means fine-tuning on a single, complete response generated by the model for each question, without any filtering based on correctness.
>
> > **Q3:** I’d like to see in-domain performance. How about using the MATH training dataset directly for training?
> >
>
> **This is an excellent question that highlights the core principle of our method.** UPFT is designed for a **self-sampling** regime, where it reinforces the model's *own* consistent reasoning patterns. When using external, gold-standard data like the MATH training set, this self-alignment property is lost.
>
> We ran this experiment. On the MATH dataset, standard SFT outperforms UPFT.
>
> | Model | Method | MATH500 |
> | --- | --- | --- |
> | Llama3.1-8B-Instruct | +SFT (MATH dataset) | **53.4** |
> | Llama3.1-8B-Instruct | +UPFT (MATH dataset) | 52.5 |
>
> This result, combined with our strong performance in the self-sampling setting below (e.g., Llama3.1 on PRM: SFT 48.4 -> **UPFT 52.0** on MATH500), confirms that UPFT's strength lies specifically in unsupervised self-improvement from noisy, model-generated data.
>
> > **Q4:** Any qualitative results? Any analysis of the samples that become solvable after fine-tuning?
> >
>
> Yes. We find that UPFT increases the model's reliability. For many problems, the base model might generate a correct answer in only 1 out of many attempts. After UPFT, the model's probability of generating the correct answer increases significantly.
>
> Take the case below as an example, for a GPQA question about optical activity, the base Qwen2.5 model answered correctly in only 1 of 16 attempts. After UPFT, it answered correctly in **3 of 16** attempts. This demonstrates that UPFT solidifies nascent correct reasoning abilities. We will add more case studies to the appendix.
>
> ```
> ***Question***
> How many of the following compounds will exhibit optical activity?
>
> (Z)-1-chloro-2-methylbut-1-ene
> (3aR,7aS,E)-8-(chloromethylene)hexahydro-4,7-methanoisobenzofuran-1,3-dione
> (2R,3S)-2,3-dimethylsuccinic acid
> (2R,3R)-2,3-dimethylsuccinic acid
> (R)-cyclohex-3-en-1-ol
> (1s,3s,5s)-cyclohexane-1,3,5-triol
> 1-cyclopentyl-3-methylbutan-1-one
>
> ***Ground_Truth***
> 3
>
> ```

---

> > ### Author Response · Authors · 2025-08-04
> >
> > Dear reviewer fBpq，
> >
> > Thank you again for your thoughtful review and feedback, which we have carefully addressed above.
> >
> > We believe these clarifications fully address your main concerns and would appreciate it if you could indicate whether they sufficiently resolve your points and might positively impact your score. If not, we welcome further discussion to better understand any remaining issues.
> >
> > Thank you again for your time and consideration.
> >
> > Best regards,
> >
> > The Authors

---

> > ### Comment · Reviewer_fBpq · 2025-08-06
> > **Response to the authors**
> >
> > Thanks for the detailed rebuttal, and apologies for the delayed response due to some eye issues.
> >
> > It seems that UPFT works well when the data is noisy and there are no gold labels. So the key aspect is the unsupervised nature, and the main baseline should be unsupervised SFT.
> >
> > Given this, SFT is not really designed for an unsupervised setting, as the name itself implies. Also, regarding SC-CoT, how many samples are used for consistency? Couldn’t simply increasing the sampling budget have a similar effect to UPFT?
> >
> > Additionally, to truly evaluate the effectiveness of unsupervised methods, I believe it would be more appropriate to compare against rl with random rewards (as mentioned in the Spurious Rewards paper) or consistency-based rewards, which have also been shown to significantly improve performance.
> >
> > Lastly, I think the claim that “+3.1 and +4.4 on strong base models are substantial” may be overstated, considering that such improvements can sometimes be achieved through better prompting alone in LLMs.

---

> ### Author Response · Authors · 2025-08-06
>
> Dear reviewer fBpq,
>
> We apologize for disturbing you and thank you again for your thoughtful review and feedback, which we have carefully addressed above.
>
> We believe these clarifications fully address your main concerns and would appreciate it if you could indicate whether they sufficiently resolve your points and might positively impact your score.
>
> As the rebuttal period is drawing to a close, please feel free to let us know if you have any further concerns or questions—we would be happy to discuss them and provide additional clarification if needed.
>
> Thank you again for your time and consideration.
>
> Best regards,
> The Authors

---

> ### Author Response · Authors · 2025-08-06
> **Reply 1/2**
>
> We sincerely thank the reviewer for their continued engagement and thoughtful follow-up questions. We are glad to hear our initial rebuttal helped clarify that **UPFT excels in the unsupervised, noisy data setting**, and that **unsupervised SFT is the key baseline**. We will address your remaining points below.
>
> ---
>
> **On the Fairness of Baselines (Unsupervised SFT and RL)**
>
> > "SFT is not really designed for an unsupervised setting... I believe it would be more appropriate to compare against rl with random rewards... or consistency-based rewards"
> >
>
> This is an excellent point about the taxonomy of methods. Our response has two parts:
>
> 1. **Why Unsupervised SFT is the most direct baseline:** You are correct that SFT stands for "Supervised" Fine-Tuning. However, applying SFT to unlabeled, model-generated data (which we term "unsupervised SFT") is a common and intuitive baseline to establish the difficulty of a problem. It answers the question: "What happens if we naively fine-tune on the model's own output?" Our results show this often fails, which is precisely what motivates our work. **UPFT is fundamentally an SFT-based method**, using the same cross-entropy loss. Therefore, comparing it to unsupervised SFT provides the most direct, apples-to-apples comparison of the training strategy itself, isolating the benefit of our prefix-tuning objective.
> 2. **Why RL is an orthogonal comparison:** RL-based methods introduce entirely different components (reward models, policy gradient optimizers like PPO) and significantly higher computational overhead. While interesting, comparing our SFT-based method to an RL-based one would be more of a comparison between training paradigms (SFT vs. RL) than an evaluation of our specific contribution. Furthermore, as noted in recent work [1, 2], the effectiveness of some RL techniques can be confounded by other factors, and simple reward schemes like random rewards have been shown to be ineffective [3]. We believe our chosen baselines provide a clearer and more direct assessment of UPFT's value.
>
> ---
>
> **On Test-Time Sampling vs. Training-Time Improvements**
>
> > "how many samples are used for consistency? Couldn’t simply increasing the sampling budget have a similar effect to UPFT?"
> >
>
> This is a crucial question that gets to the heart of what UPFT accomplishes.
>
> **UPFT is a training-time algorithm, while self-consistency (SC) is a test-time decoding strategy. They are complementary, not competing.** All our experiments, for both baselines and UPFT, used a standard SC-CoT setup with N=4 samples to ensure a fair comparison.
>
> The key difference is this:
>
> - **Increasing the test-time sampling budget** (e.g., from N=4 to N=32) gives a model more chances to find a correct answer *that it could already generate with some non-zero probability*. It doesn't make the model fundamentally better.
> - **UPFT, as a training method, improves the model itself.** It increases the base probability of generating correct reasoning steps. This means that *each* sample drawn at test time is more likely to be correct. As our results show, UPFT boosts performance at N=4 and N=32, demonstrating it makes the model more reliable, which is a more fundamental improvement than simply sampling more.

---

> ### Author Response · Authors · 2025-08-07
> **Reply 2/2**
>
> **On the Significance of Our Gains (vs. Prompt Engineering)**
>
> > "I think the claim that “+3.1 and +4.4 on strong base models are substantial” may be overstated, considering that such improvements can sometimes be achieved through better prompting alone"
> >
>
> We understand the skepticism, as prompting can indeed have a large effect. We argue our gains are substantial for three reasons, including a new experiment to address this directly.
>
> 1. **We ran a new prompt engineering experiment.** To test your hypothesis, we used Deepseek-v3 to generate four diverse, high-quality system prompts for the Qwen2.5-Math-7B-Instruct model. We evaluated them against the model's default prompt on the challenging AIME24 and GPQA benchmarks. **The results below show that the default prompt is already optimal, and that prompt engineering does not easily yield further gains on these hard tasks.**
>
>
>     | System Prompt | AIME24 Acc@8 | GPQA Acc@8 |
>     | --- | --- | --- |
>     | **0 (Default - Our Setting)** | **15.8** | **10.1** |
>     | 1 (Generated by DS-v3) | 12.3 | 7.9 |
>     | 2 (Generated by DS-v3) | 13.8 | 9.1 |
>     | 3 (Generated by DS-v3) | 14.6 | 9.3 |
>     | 4 (Generated by DS-v3) | 13.7 | 8.8 |
>
>     This experiment confirms that our reported improvements come from our method, not from a suboptimal prompt.
>
> 2. **Gains on challenging benchmarks are hard-won.** As literature confirms [5, 6], the effectiveness of prompt engineering diminishes on complex, distribution-shifted data like AIME24 and GPQA. Achieving a **+3 to +4 point absolute improvement** on these benchmarks through a training modification is a significant result.
> 3. **Our gains are consistent across multiple model families.** We demonstrated UPFT's effectiveness on Llama3.1, Qwen2.5, and DeepSeek-R1 models. This consistency, especially on models like Llama that are less prone to training/test data contamination [1, 2], shows that UPFT provides a generalizable improvement to reasoning, not a model-specific artifact.
>
> We thank you again for the rigorous and constructive feedback. We are confident that our method provides a practical and significant contribution to unsupervised model improvement, and we will integrate these clarifications and new results into the final paper.
>
> **References**
>
> [1] Chandak et al. Incorrect Baseline Evaluations Call into Question Recent LLM-RL Claims. Notion, 2025.
>
> [2] Wu et al. REASONING OR MEMORIZATION? UNRELIABLE RE-SULTS OF REINFORCEMENT LEARNING DUE TO DATACONTAMINATION. arxiv, 2025.
>
> [3] Li et al. PRESERVING DIVERSITY IN SUPERVISED FINE-TUNING OF LARGE LANGUAGE MODELS. ICLR, 2025.
>
> [4] Shao et al. Spurious Rewards: Rethinking Training Signals in RLVR. arxiv, 2025.
>
> [5] Li et al. Robust Prompt Optimization for Large Language Models Against Distribution Shifts. EMNLP, 2023.
>
> [6] Ghosh et al. A Closer Look at the Limitations of Instruction Tuning. ICML, 2024.

---

> ### Author Response · Authors · 2025-08-07
>
> Dear Reviewer fBpq,
>
> As the rebuttal period is drawing to a close, please let us know if you have any remaining concerns or questions that you feel we have not addressed, and which might affect your judgment of our paper's value.
>
> We would be more than happy to discuss them further and provide additional clarifications as needed. We believe this would be very helpful for improving the quality of our paper.
>
> If all your concerns have been adequately addressed, please kindly consider updating the score to reflect your current assessment. We appreciate your time and consideration!
>
> Best regards,
> The Authors

---

### Official Review · Reviewer_JFVJ · 2025-07-03

**Clarity:** 3
**Significance:** 3
**Originality:** 3
**Rating:** 5
**Confidence:** 4

**Summary:**

This paper proposes a novel self-improvement method for LLMs that enhances mathematical reasoning by training on only a small number of top-k examples. The approach not only reduces sampling and training costs but also outperforms rejection sampling fine-tuning in terms of performance.

**Questions:**

- Can the proposed method generalize to tasks beyond mathematical reasoning, such as code generation?
- Is there a performance difference when applying UPFT to models trained with long vs. short CoT?
- If responses are distilled from a stronger model to fine-tune a weaker one, is training on only the first few tokens still important?

**Ethical Concerns:**

["NO or VERY MINOR ethics concerns only"]

**Final Justification:**

I have carefully read the responses and the other reviews. I decide to retain my score.

**Limitations:**

yes

**Quality:**

3

**Strengths And Weaknesses:**

### Strengths:
- This paper presents a novel observation and empirical analysis of the prefix self-consistency phenomenon.

- Based on this insight, the authors design UPFT, an efficient SFT (supervised fine-tuning) method for self-improvement of LLMs.

- The proposed UPFT method outperforms existing approaches such as RFT on mathematical reasoning tasks, while significantly reducing sampling and training costs.

- The authors also provide theoretical analysis to explain why UPFT is effective.

- Experiments across LLMs with varying levels of mathematical ability show that UPFT consistently leads to performance improvements.

### Weaknesses:
- The experimental evaluation is limited to mathematical reasoning tasks, which raises questions about the generalizability of the proposed method to other domains.

- While the title “The First Few Tokens Are All You Need” may overstate the claim. In practice, the SFT procedure still involves structure tuning with the full reasoning trace, suggesting that early tokens are important but not sufficient on their own.

- There appears to be a potential typo in the notation for rejection sampling on line 108, which may cause confusion.

---

> ### Author Rebuttal · Authors · 2025-07-31
>
> We sincerely thank you for your thoughtful review and recognition of our work's novelty and efficiency. Your constructive feedback is very helpful. We address your questions below.
>
> > **W1&Q1:** The experimental evaluation is limited to mathematical reasoning tasks, which raises questions about the generalizability of the proposed method to other domains.
> >
>
> Yes, our method shows strong generalization to non-mathematical domains.** We'd like to highlight a key result that may have been missed:
>
> - **Our evaluation already includes a non-math, cross-domain benchmark.** The **GPQA** dataset, as described in the paper, consists of PhD-level questions across **biology, physics, and chemistry**, not math.
> - **UPFT demonstrates strong Out-of-Distribution (OOD) generalization.** Our models were trained *exclusively* on mathematical reasoning data. The consistent performance improvements on GPQA (e.g., **+4.6** for DeepSeek on LIMO, see Table in response to R_fBpq) show that UPFT enhances a general reasoning capability that transfers OOD.
>
> While we did not test on code generation, the strong OOD results on GPQA suggest UPFT is not limited to a single domain and could be promising for other reasoning tasks like code generation, which we leave for future work.
>
> > **W2:** While the title “The First Few Tokens Are All You Need” may overstate the claim. In practice, the SFT procedure still involves structure tuning with the full reasoning trace, suggesting that early tokens are important but not sufficient on their own.
> >
>
> Thank you for this valuable suggestion. Our title was inspired by "Attention is All You Need" to be memorable, but we agree it could be interpreted more literally than intended. We acknowledge that the small percentage of full-trace tuning is important for preserving the model's output structure. We will revise the title to be more precise in the final version, e.g.. “**The First Few Tokens Matter**”.
>
> > **Q2:** Is there a performance difference when applying UPFT to models trained with long vs. short CoT?
> >
>
> **Yes, and our results suggest UPFT is broadly effective regardless of the CoT length of the training data.** As shown in Table 2, the training datasets have vastly different average CoT lengths (e.g., LIMO at 491.8 vs. U-Hard at 393.3 for Qwen2.5). Despite these differences, UPFT consistently delivers gains:
>
> - **Qwen2.5 on LIMO (long CoT):** Base 51.4 -> UPFT 52.5 (**+1.1** avg.)
> - **Qwen2.5 on U-Hard (shorter CoT):** Base 51.4 -> UPFT 54.5 (**+3.1** avg.)
>
> This indicates that UPFT's benefit, which comes from reinforcing correct initial reasoning steps, is not dependent on the length of the full reasoning traces in the source data.
>
> > **Q3:** If responses are distilled from a stronger model to fine-tune a weaker one, is training on only the first few tokens still important?
> >
>
>  **Not necessarily. Our method's core assumption is that we are fine-tuning on the model's *own* generated samples.** This self-sampling is key to reinforcing the model's inherent, correct reasoning pathways. Distilling from a stronger, external model is a different setting where this assumption does not hold.
>
> To verify this, we ran a new experiment where we fine-tuned Llama3.1-8B-Instruct but generated the prefixes from stronger models.
>
> | **Decoding Model** (Tuned) | **Prefix Generation Model** (External) | **GSM8K** | **MATH500** |
> | --- | --- | --- | --- |
> | Llama3.1-8B-Instruct | Llama3.1-8B-Instruct (Self-Generated) | **82.0** | **51.0** |
> | Llama3.1-8B-Instruct | Llama3.1-70B-Instruct (Stronger) | 80.3 (↓1.7) | 49.6 (↓1.4) |
> | Llama3.1-8B-Instruct | Qwen2.5-Math-72B-Instruct (Stronger) | 78.5 (↓3.5) | 44.8 (↓6.2) |
>
> As shown, performance **degrades** when using prefixes from an external model, even a much stronger one. This confirms that UPFT is most effective in the self-improvement setting, where it aligns the model's distribution with its own most consistent reasoning starts.
>
> > **W3:** Typo about line 108.
> >
>
> Thank you for catching this. We will correct it in the final version.

---

> ### Author Response · Authors · 2025-08-04
>
> Dear reviewer JFVJ，
>
> Thank you again for your thoughtful review and feedback, which we have carefully addressed above.
>
> We believe these clarifications fully address your main concerns and would appreciate it if you could indicate whether they sufficiently resolve your points and might positively impact your score. If not, we welcome further discussion to better understand any remaining issues.
>
> Thank you again for your time and consideration.
>
> Best regards,
>
> The Authors

---

> > ### Comment · Reviewer_JFVJ · 2025-08-07
> > **Reviewer JFVJ Comments**
> >
> > Thank you for your clarifications, I have no more questions.

---

> > > ### Author Response · Authors · 2025-08-07
> > >
> > > Thank you for your consideration and for helping us improve the quality of our paper! As the rebuttal period is drawing to a close, please let us know if you have any remaining concerns or questions that you feel we have not addressed.

---

### Official Review · Reviewer_XozN · 2025-07-03

**Clarity:** 4
**Significance:** 4
**Originality:** 3
**Rating:** 5
**Confidence:** 4

**Summary:**

This paper proposes the Unsupervised Prefix Fine-Tuning (UPFT), a method for improving the efficiency of LLM reasoning by fine-tuning models on minimal prefixes (shared reasoning phase). UPFT significantly reduces both training and sampling time, while achieving competitive performance compared to the baseline method, Rejection Sampling Fine-Tuning (RFT).

**Questions:**

n/a

**Ethical Concerns:**

["NO or VERY MINOR ethics concerns only"]

**Final Justification:**

I have read the authors' responses and the other reviews. I will retain my score to show my support for this paper.

**Limitations:**

yes

**Quality:**

4

**Strengths And Weaknesses:**

**Strengths**

-	This paper identifies a key insight, Prefix Self-Consistency— different solution paths often share a common initial reasoning phase—which is intriguing, and the proposed method UPFT is both computationally efficient and reduces the need for data labeling or rejection sampling.
-	The paper is well-organized, beginning with an empirical observation that early reasoning steps are highly consistent across reasoning trajectories. This is followed by a theoretical analysis from a Bayesian perspective and the introduction of the Unsupervised Prefix Fine-Tuning method, which aims to maximize reasoning trace coverage while maintaining high accuracy. The writing is clear, and the paper is easy to follow.
-	Experiments are comprehensive, involving three models (Llama-3.1-8B-Instruct, Qwen2.5-Math-7B-Instruct, DeepSeek-R1-Distill-Qwen-7B), evaluated on four benchmarks (GSM8K, MATH500, AIME24, GPQA Diamond), and the method is compared against the baseline in both unsupervised sampling and supervised sampling settings.

**Weaknesses**

This paper makes clear contributions to both methodology and empirical evaluation. I don’t see any major weaknesses.

---

> ### Author Rebuttal · Authors · 2025-07-31
>
> We want to express our sincere gratitude for your positive and encouraging review. We are thrilled that you found our paper "excellent" and our contributions clear and significant. Your understanding of our core insight and appreciation for our comprehensive experiments are very motivating. We kindly ask for your continued support during the internal discussion phase. Your endorsement will be invaluable in contextualizing our contributions for the other reviewers and the Area Chair.

---

### Decision · Program_Chairs · 2025-09-17

**Decision:**

Accept (poster)

**Comment:**

(a) Summary of claims and findings:
This paper introduces Unsupervised Prefix Fine-Tuning (UPFT), a novel and highly efficient method for improving the reasoning capabilities of Large Language Models (LLMs). The core insight, termed "Prefix Self-Consistency," is the empirical observation that diverse, correct reasoning trajectories for a given problem often share the same initial sequence of tokens. Leveraging this, UPFT fine-tunes the model on just these short initial prefixes (e.g., the first 8 tokens) of model-generated solutions. Crucially, this is done in a fully unsupervised manner, using a single unfiltered response per question without requiring ground-truth labels or expensive rejection sampling to filter for correct answers. The authors claim that this simple technique can match the performance of costly supervised methods like Rejection Sampling Fine-Tuning (RFT) while reducing sampling costs by over 99% and training time by over 75%.

(b) What are the strengths of the paper?
The paper's primary strengths are its novelty, simplicity, and remarkable efficiency.

Novel and Intuitive Idea: The core concept of "Prefix Self-Consistency" is a simple yet powerful insight into the behavior of LLMs during reasoning. The idea of reinforcing these stable initial steps is elegant and well-motivated.

High Efficiency and Practicality: The method's main appeal is its extreme resource efficiency. By avoiding the need for labeled data and expensive sampling/filtering pipelines (like RFT), and by training on very short sequences, UPFT dramatically lowers the barrier to improving LLM reasoning, making it accessible to those with limited computational resources.

Strong Empirical Results: The authors conduct a comprehensive set of experiments across multiple models (Llama, Qwen, DeepSeek) and challenging reasoning benchmarks (GSM8K, MATH, AIME, GPQA). The results compellingly show that UPFT consistently improves performance over the base model, whereas naive unsupervised fine-tuning (SFT) often leads to performance degradation.

Excellent Rebuttal: The authors provided an exceptionally thorough and convincing rebuttal, running several new experiments (multi-seed validation, weaker model backbones, training time analysis, prompt engineering baselines) that substantially strengthened the paper's claims and addressed most of the reviewers' concerns.

(c) What are the weaknesses of the paper? What might be missing in the submission?
The primary weaknesses raised during the review process revolved around the perceived magnitude of the performance gains and the generalizability of the method.

Significance of Performance Gains: One reviewer (fBpq) argued that the absolute performance improvements were "marginal" and questioned the practical utility. While the gains are not massive in all cases, this critique somewhat overlooks the challenging unsupervised setting where the most direct baseline (unsupervised SFT) often fails entirely.

Generalizability: A concern was raised (JFVJ) that the experiments were overly focused on mathematical reasoning. The authors effectively countered this by highlighting the strong out-of-distribution performance on the GPQA dataset, which covers biology, physics, and chemistry.

Initial Clarity: Some aspects of the initial submission were unclear, leading to significant misunderstandings by at least one reviewer (H5bo) regarding the unsupervised nature of the method and the interpretation of an ablation study. However, the authors have committed to clarifying these points in the final version.

(d) Provide the most important reasons for your decision to accept/reject.
I recommend acceptance because the paper presents a well-motivated, simple, and highly efficient method for a significant problem. The strengths clearly outweigh the weaknesses.

The core reason for my recommendation is that UPFT successfully addresses the challenging task of model self-improvement in a fully unsupervised and resource-scarce setting. The most direct and fair baseline in this scenario is standard SFT on the model's own noisy outputs, which the paper shows is unreliable and often harmful. UPFT's ability to deliver consistent and statistically significant performance gains in this exact setting is a strong and valuable contribution.

While two reviewers recommended rejection, their concerns were either thoroughly addressed by the authors' rebuttal or stemmed from fundamental misunderstandings. Reviewer H5bo's critique was predicated on a factually incorrect assumption that the method used ground-truth labels and a non-standard definition of "unsupervised learning." Reviewer fBpq's skepticism about the practical utility, while a valid perspective, was countered by new experiments from the authors, including multi-seed runs demonstrating statistical significance and a prompt engineering baseline showing the gains were not trivial. The two positive reviewers (XozN, JFVJ) correctly identified the novelty and significance of the work. Given the evidence, especially after the extensive rebuttal, the paper stands as a solid, well-executed piece of research.

(e) Summarize the discussion and changes during the rebuttal period.
The rebuttal period was highly effective and substantially improved the paper's standing. The discussion centered on three main axes:

Clarifying "Unsupervised": Reviewer H5bo fundamentally misunderstood the method, believing it used ground-truth answers. The authors clarified that their method is fully unsupervised, training on single, unfiltered model outputs. This misunderstanding, which the authors flagged in a confidential comment, was a major source of the initial negative rating.

Demonstrating Robustness and Significance: In response to concerns from Reviewer fBpq about statistical significance and Reviewer H5bo about generalization to weaker models, the authors ran extensive new experiments. They provided results over 5 random seeds (confirming statistical significance with p<0.01), evaluated UPFT on smaller 1B and 3B models (showing consistent gains), and added a comprehensive training time analysis (demonstrating a >70% time saving). This new evidence directly addressed the reviewers' core empirical concerns.

Contextualizing Performance Gains: Reviewer fBpq remained skeptical about the value of the gains. The authors successfully argued that in the context of unsupervised learning from noisy data (where SFT degrades performance), UPFT's consistent improvements are significant. They further ran a new experiment showing that simple prompt engineering could not replicate the gains, reinforcing that the improvement comes from their method.

In summary, the authors successfully corrected factual misunderstandings, provided substantial new evidence to back their claims, and agreed to minor changes like clarifying the text and toning down the title. The rebuttal successfully convinced Reviewer JFVJ and provided strong counter-arguments to the points raised by Reviewers fBpq and H5bo.